# The Influence of Bio-Stimulants and Foliar Fertilizers on Yield, Plant Features, and the Level of Soil Biochemical Activity in White Lupine (*Lupinus albus* L.) Cultivation

**Alicja Niewiadomska [1], Hanna Sulewska [2], Agnieszka Wolna-Maruwka [1], Karolina Ratajczak [2],*, Zyta Waraczewska [1] and Anna Budka [3]**

[1] Department of General and Environmental Microbiology, Poznań University of Life Sciences, ul. Szydłowska 50, 60-656 Poznań, Poland; alicja.niewiadomska@up.poznan.pl (A.N.); amaruwka@up.poznan.pl (A.W.-M.); zyta.waraczewska@wp.pl (Z.W.)

[2] Department of Agronomy, Poznan University of Life Sciences, ul. Dojazd 11, 60-632 Poznań, Poland; hanna.sulewska@up.poznan.pl

[3] Department of Mathematical and Statistical Methods, Poznań University of Life Sciences, ul. Wojska Polskiego 28, 60-637 Poznań, Poland; anna.budka@up.poznan.pl

* Correspondence: karolina.ratajczak@up.poznan.pl

**Abstract:** The aim of this study is to assess the effect of two biostimulators (Titanit, Rooter) and six foliar fertilizers (Optysil, Metalosate Potassium, Bolero Bo, ADOB 2.0 Zn IDHA, ADOB B, ADOB 2.0 Mo) on white lupine. In addition, we evaluated the enzymatic activity of dehydrogenase, acid, and alkaline phosphatases, catalase, the level of biological nitrogen fixation, yield, plant biometric, chlorophyll fluorescence and chlorophyll content. A field experiment was conducted between 2016 and 2018 at the Gorzyń Experimental and Educational Station, Poznań University of Life Sciences in Poland. The best effects in plant yield were obtained after the application of Optysil or ADOB Zn IDHA. The three years results of dehydrogenase (DHA), alkaline phosphatase (PAL), and the biological index of soil fertility (BIF), show that the bio-stimulants and most of the foliar fertilizers used did not always stimulate the activity of these enzymes and index in the white lupine crops, as compared with the control treatment. Analysis of the results of the acid phosphatase activity (PAC) shows that during the entire white lupine growing season the foliar fertilizers and bio-stimulants decreased the activity of this enzyme. This effect was not observed when the Metalosate potassium foliar fertilizer was applied. The field analyses of biological nitrogen fixation showed that the fertilizers and bio-stimulants significantly stimulated nitrogenase activity under the white lupine plantation. The best effects in plant yield were obtained after application Optysil or ADOB Zn IDHA.

**Keywords:** soil enzymatic activity; biological index fertility; nitrogenase activity; microelements fertilization (Ti, Si, B, Mo, Zn)

## 1. Introduction

The degradation of the soil environment, excessive use of chemicals, depletion of natural resources, as well as the decreasing biodiversity instigated the European Union to make a decision about the need for integrated crop cultivation and protection [1]. Since 2014 the recommendations concerning integrated protection and cultivation have been in force in Poland. At present we can see the transitional phase between conventional and sustainable agriculture. In order to meet the assumptions of sustainable agriculture it is necessary to diversify the crop structure and minimize the excessive

share of cereals. It is also necessary to use integrated methods of agricultural production, so it might be particularly important to restore legume plantations [2].

The significance of legumes in sustainable agriculture is increasing because they improve the physicochemical properties of soil, increase the content of organic matter by leaving large quantities of crop residues, and reduce the need to apply nitrogen fertilizers. White lupine (*Lupinus albus* L.) is one of the most important crops in this group of plants in Poland. It has been the longest known crop species of the *Lupinus* genus. Because of its very high content of protein and fat, especially in seeds, it has been used for human nutrition for thousands of years, despite its high content of bitter alkaloids [3]. It was only in 1930 that low-alkaloid forms were obtained. Because of the introduction of new varieties, the cultivation of white lupine with low alkaloid content became popular in Poland. Between 2005 and 2015 the area of cultivation of large-seeded legumes increased almost four times so that in 2015 they covered an area of 407,000 ha [4].

Lupine species have the largest share in this group of crops. On the other hand, the area of plantations with small-seeded legumes, such as clover and alfalfa, did not fluctuate much in that decade and in 2015 they covered an area of 93,000 ha [5].

Legumes are characterized by the ability to coexist with the nitrogen-fixing diazotrophic bacteria (*Rhizobium*). In order to increase the protein content in plants, which depends on the system developed by the plant and rhizobia, it is necessary to find agents improving the efficiency of this symbiosis.

Scientists are more and more interested in bio-stimulants, which are defined as materials containing one or more active substances and/or microorganisms. They improve the uptake of nutrients by plants, their tolerance to abiotic and biotic stress, and the quality of crops [6]. Bio-stimulants also increase the activity of rhizosphere microorganisms and soil enzymes, as well as they stimulate hormone production and photosynthesis [7]. They also promote the overall plant growth, including increased biomass and crop yields [8]. In the group of synthetic bio-stimulants, there are preparations containing growth regulators, phenolic compounds, inorganic salts, and beneficial nutrients [9,10], which naturally occur in plants in trace amounts (e.g., titanium and silicon). They act mainly by the stimulation of numerous physiological processes, which has a positive effect on plant yield and crop quality. Nutrients assimilable by plants, reduces the impact of stress, which affects the growth and development of plants. They regulate the uptake of macro- and microelements, alleviate the negative effects of periodic water shortage, high salinity, as well as activates the natural immune mechanisms of plants. They also strengthen cell walls and reduce the susceptibility of plants to mechanical damage [11]. Microelements regulate biochemical processes occurring in plants, being part of most enzymes or acting as their activators, therefore their deficit may lead to the inhibition of specific enzymatic reactions, which in turn leads to disorders of many biochemical and physiological processes, adversely affecting the growth and plant development [12,13]. There are many fertilizers that are enriched with amino acids, organic compounds, or surfactants. For example, potassium in fertilizer is in the form of very small molecules complexed with a unique set of natural amino acids. In turn, boron in the fertilizer is in the form of sodium pentaborate decahydrate, and the addition of sorbitol ensures rapid uptake of the fertilizer through the leaves of fertilized plants and high efficiency of the fertilizer. Zinc in modern fertilizers is chelated with the biodegradable IDHA chelating agent, because of which it also gains a form that is very well absorbed by plants. This fertilizer increases the plants' resistance to drought and diseases and increases the germination of seeds. It is produced in the form of microgranules, based on modern microgranulation technology. The manufacturer of molybdenum fertilizer has developed a liquid formula of the fertilizer additionally enriched with biodegradable tensides, which decreases the surface tension of the working liquid and increases the efficiency of covering the leaf blade during spraying increases [14].

Essential plant nutrients are mainly applied to soil and plant foliage in order to achieve maximum economic yields. Soil application is more common and most effective for nutrients that are required in high quantities. However, under certain circumstances, foliar fertilization is more economic and effective. Because of the intensified cultivation foliar fertilization has become an indispensable

agrotechnical procedure. Plants exhibit the highest demand for potassium and nitrogen (more than 200 kg in terms of the yield per 1 ha), and the lowest demand for zinc, boron, copper, and molybdenum. Plants need only a few grams of molybdenum in terms of the yield per hectare. This means that foliar fertilization is particularly recommended and effective when it is necessary to supply micronutrients to crops [15].

Each agrotechnical treatment, i.e., the use of fertilizers or bio-stimulants, may cause changes in the soil environment. There have been numerous studies showing various effects of these treatments on the count of selected groups of microorganisms and the amount of soil enzymes they secrete [16].

Measurement of the activity of soil enzymes provides information about the quality of soil. This procedure is important as it indicates the biochemical activity of soil. Enzymes are thought to be good and sensitive indicators because they quickly react to changes in soil caused by natural and anthropogenic factors. Apart from that, it is easy to measure their activity, which affects the main microbiological reactions involving the cycles of nutrients in soil. Studies also showed that agrotechnical procedures influence the enzymatic activity more than other biochemical parameters [17].

The aim of this study is to assess the effect of selected bio-stimulants (Tytanit, Rooter) and foliar fertilizers (Optysil, Metalosate potassium, Bolero Bo, ADOB 2.0 Zn IDHA, ADOB B, ADOB 2.0 Mo) on the yield and plant features, activity of dehydrogenase, acid and alkaline phosphatases, and catalase, as well as the level of biological nitrogen fixation based on the activity of nitrogenase in a white lupine plantation.

## 2. Material and Methods

### 2.1. Experimental Design

A field experiment was conducted between 2016 and 2018 at the Gorzyń Experimental and Educational Station, Poznań University of Life Sciences. The GPS coordinates of the experiment are as follows: N-52.56589, E-015.90556, 65 m AMSL. Each year one-factor experiment was conducted as randomized block design in four replications with the following nine factor levels: 1. control treatment—plants not treated with preparations; 2. Tytanit; 3. Optysil; 4. Metalosate Potassium; 5. Rooter; 6. Bolero Mo; 7. ADOB Zn IDHA; 8. ADOB B; 9. ADOB 2.0 Mo. Each fertilizer was applied in a timely manner, according to the manufacturer's recommendations (Table 1).

An experiment was conducted on white lupine (*Lupinus albus* L.) of the Butan cultivar. The lupine seeds were inoculated with the effective strain of *Bradyrhizobium lupinus* root nodule bacteria directly before sowing by using nitragina. Nitragina is a single-component graft, containing a specific bacterial strain for a specific legume plant, in which peat is a carrier. The Butan cultivar can be grown all over Poland, this variety is insensitive to delayed sowing; its growing period is 2–14 days shorter than that of traditional varieties and it is less susceptible to diseases caused by *Fusarium* fungi. The cultivar is more valuable as a feed and it has high content of protein (32–37%) and fat (10–12%), while the content of alkaloids is about 30–40% lower.

The seeds were sown (4 April 2016, 4 April 2017 and 7 April 2018) on plots with an area of 21 m$^2$, with a distance of rows of 15 cm, and sowing density of 75 seeds per 1 m$^2$.

According to the FAO/WRB classification [18], the soil in the experimental plots is a typical lessive soil formed from light loamy sands, deposited in a shallow layer on light loam (*Haplic Luvisols*) (Table 2). The soil texture was determined by means of a sieve (sand fraction) for the silt and clay fraction [19].

The agrotechnical and cultivation treatments were carried out in accordance with the principles of good agricultural and experimental practice for this species [20]. In the autumn before winter plowing, basic macronutrients were supplied to the soil in the form of multi-component fertilizer Polifoska 4 in the amount of 350 kg ha$^{-1}$ (N—4%, P—12%, K—32%). Before sowing, urea in the amount of 30 kg ha$^{-1}$ was used.

**Table 1.** The terms and doses of bio-stimulants and fertilizers applied in the experiment.

| Bio-Stimulants/Foliar Fertilizers | | Term and Dose of Bio-Stimulant | Fertilizer Characteristics |
|---|---|---|---|
| Bio-stimulants | Tytanit | I: leaf and shoot development (BBCH 13–29)—0.3 dm$^3$ ha$^{-1}$ II: inflorescence development (BBCH 51–59)—0.3 dm$^3$ ha$^{-1}$ III: beginning of pod development (BBCH 71)—0.3 dm$^3$ ha$^{-1}$ | Liquid, mineral stimulant containing titanium (Ti). It increases the yield volume and development of plants, improves yield quality parameters and increases plants' natural resistance to stress factors. Composition: 8.5 g Ti (dm$^3$)$^{-1}$ |
| | Rooter | BBCH 13–14—1 dm$^3$ ha$^{-1}$ | Bio-stimulant—it stimulates the growth of the root system, accelerates regeneration and improves the uptake of soil minerals. Composition: P$_2$O$_5$ 13.0%; K$_2$O 5.0% |
| Foliar fertilizers | Optysil | I: leaf and shoot development (BBCH 15–29)—0.5 dm$^3$ ha$^{-1}$ II: inflorescence development (BBCH 51–55)—0.5 dm$^3$ ha$^{-1}$ III: beginning of pod development (BBCH 71–73)—0.5 dm$^3$ ha$^{-1}$ | Liquid, silicon antistressor stimulating the growth and development of plants, activating their natural immune systems and increasing tolerance to unfavorable cultivation conditions. Composition: 200 g SiO$_2$ (dm$^3$)$^{-1}$ |
| | Metalosate Potassium | 2–3 treatments every 10–14 days during intensive growth—3 dm$^3$ ha$^{-1}$ | Liquid foliar fertilizer containing an easily absorbable form of potassium, which supplements potassium deficit in plants with amino acids. Composition: K$_2$O 24% |
| | Bolero Mo | Before florescence—1.5 dm$^3$ ha$^{-1}$ | Liquid foliar fertilizer containing boron and molybdenum to supplement the deficit of these elements in plants. Composition: B 8.2%; Mo 0.8% |
| | ADOB 2.0 Zn IDHA | Before florescence—1 dm$^3$ ha$^{-1}$ | Foliar fertilizer containing zinc (Zn) fully chelated by biodegradable chelating agent IDHA. Composition: Zn 100 g kg$^{-1}$ (weight percentage content 10, chelated by IDHA) |
| | ADOB B | I: before florescence—2 dm$^3$ ha$^{-1}$ II: after florescence on pods—1 dm$^3$ ha$^{-1}$ | Liquid, highly concentrated foliar fertilizer containing boron that regulates auxin activity and participates in cell division. Composition: N 78 g kg$^{-1}$; B 150 g kg$^{-1}$ |
| | ADOB 2.0 Mo | early stages of development—0.15 dm$^3$ ha$^{-1}$ | Liquid, single-component fertilizer which increases the rate and efficiency of use of nitrogen by plants and improves interaction with iron. Composition: Mo 20% |

**Table 2.** The texture of soil sampled at a depth of 0–25 cm and the soil chemical properties of the 3-year experiment.

| | Percentage of Soil Fractions | | | Texture Class |
|---|---|---|---|---|
| Fraction [mm] | Sand 2–0.05 | Silt 0.05–0.002 | Clay <0.002 | |
| | 78 | 18 | 4 | LS |
| **Soil Chemical Properties** | | | | |
| pH in 1 mol KCl | | 6.0 | | |
| Phosphorus P (mg·kg$^{-1}$) | | 70.1 | | |
| Potassium K (mg·kg$^{-1}$) | | 99.3 | | |
| Magnesium Mg (mg·kg$^{-1}$) | | 56.7 | | |
| Manganese Mn (mg·kg$^{-1}$) | | 303.4 | | |
| Zinc Zn (mg·kg$^{-1}$) | | 10.9 | | |
| Copper Cu (mg·kg$^{-1}$) | | 2.6 | | |
| Iron Fe (mg·kg$^{-1}$) | | 1525.2 | | |
| Boron B (mg·kg$^{-1}$) | | >20 | | |
| Organic carbon (%) | | 0.5 | | |
| Percent of caries | | 0.8 | | |

LS—loamy sand.

The agrotechnical procedures were carried out in accordance with the rules adopted for the species used in the test. White lupine was sown in early April. The following products were used for weed control: Afalon Dispersive 450 EC (1.1 L ha$^{-1}$) in April, Basagran 480 SL (2.6 L ha$^{-1}$) and Betanal MaxPro 209 OD (1.25 L ha$^{-1}$) in May. Fusilade Forte 150 EC (1.0 L ha$^{-1}$) was additionally applied in June. The following products were sprayed to protect the plants from diseases: Gwarant 500 SC (2.0 L ha$^{-1}$) in May and Korazzo 250 SC (1.0 L ha$^{-1}$) in mid- and late June.

*2.2. Weather Conditions*

During the growing seasons in 2016 and 2017 the weather conditions were similar in terms of temperature and rainfall. During the growing season the highest average air temperature was noted in July both in 2016 (19.5 °C) and 2017 (18.9 °C), whereas the lowest temperature was noted in April, i.e., 8.7 °C in 2016 and 7.5 °C in 2017. However, the weather conditions in 2018 were different than in the previous years (Figure 1). The highest average temperature was noted in August (21.2 °C), whereas the lowest was noted in May (12.7 °C). As far as the average monthly temperature from April to September is concerned, 2018 was the warmest—it was 2.9 °C warmer than 2016 and 1.7 °C warmer than 2017. In 2016 there was drought only at the end of the growing season. Likewise, in 2017 there was no rainfall deficit. On the contrary, it was a wet year, especially from June to August. On the other hand, in 2018 rainfall was unevenly distributed and there were droughts that were particularly unfavorable for plants in May, June, and August.

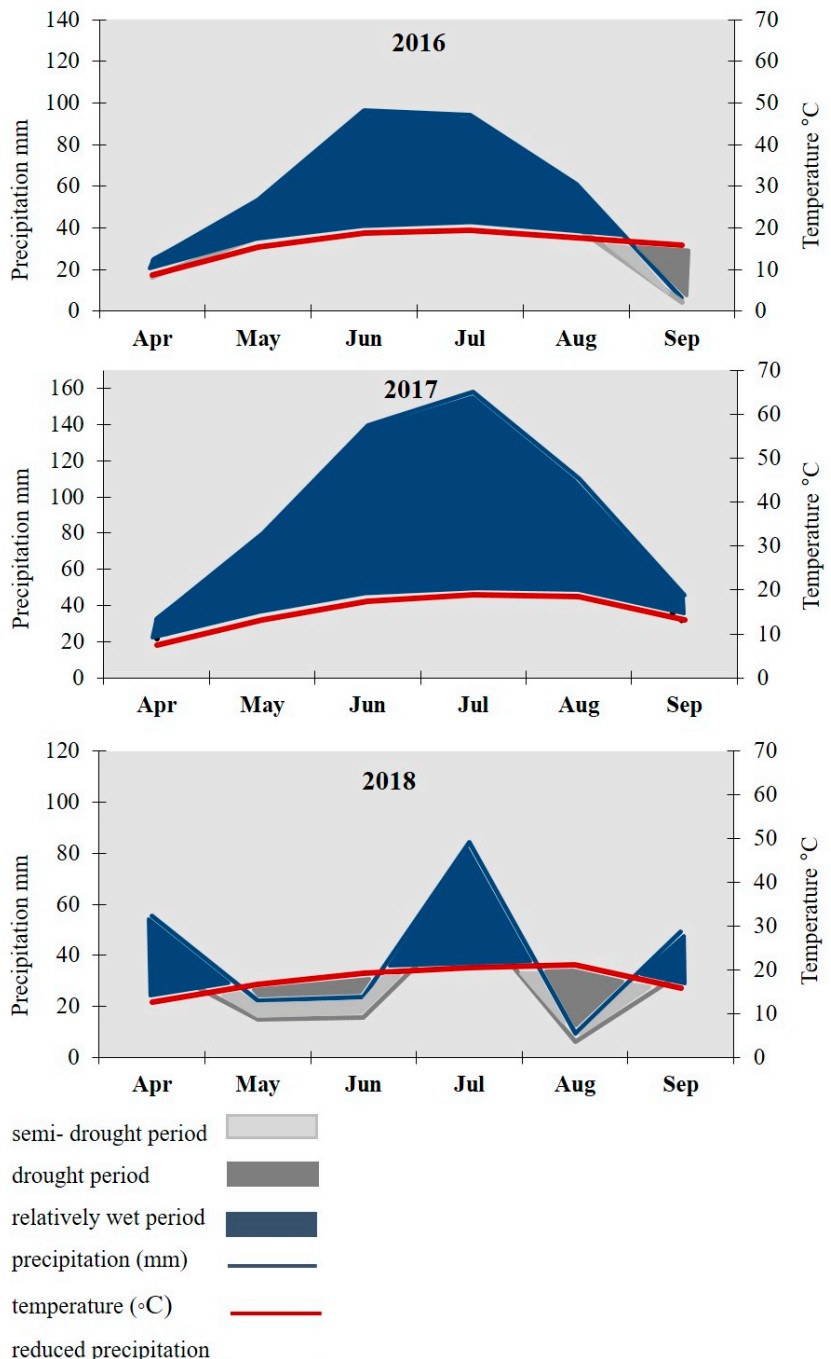

**Figure 1.** Climate graphs according to Walter [21] characterizing weather conditions in Gorzyń.

### 2.3. Influence of Fertilizers on Nitrogenase Activity (Diazotrophy)

Nitrogenase activity was estimated using the acetylene reduction assay (ARA) at the beginning of the plants' flowering [22]. For this purpose, five plants were randomly selected in plots, in a given experimental treatment and directly were placed tightly in sealed test vials (2000 mL) at the field, purified $C_2H_4$ was injected to obtain an acetylene concentration of 10% (*v/v*) in the gas phase (air). After an hour, 1 mL of the gas phase was taken from inside of the test vessels with a Hamilton gas-tight syringe and stored in small glass vials, which were sealed with rubber septa and aluminum seals. Ethylene concentration was determined using gas chromatograph CHROM 5 (Laboratorni Přistroje, Praha, Czech Republic, 1980). Nitrogenase activity was determined based on the quantity of acetylene

reduced to ethylene and expressed in $nmolC_2H_4$ produced per plant per hour ($nMC_2H_4$ $plant^{-1}$ $h^{-1}$). The results are the mean value of five replications from each measurement.

## 2.4. Plant Biometric Assessment

The plant height (from soil surface to the highest plant point) and number of pods per plant were measured. Shoot, root, and nodule dry mass were determined after drying for 2 days at 70 °C until reaching constant weight. All the biometric traits were measured on 10 randomly selected plants from each object and replication during plant vegetation and before harvest. The total one-sided area of leaf per unit ground surface area expressed by the leaf area index (LAI) and was measured at three randomly selected places of each plot at the BBCH stage 69 using a SunScan Canopy Analysis System type SSI. Lupine was harvested at one stage (BBCH 90–92) with a 1.35-m wide plot combine. The yield of clean seeds was determined in $dt \cdot ha^{-1}$, given at a standardized (15%) water content and thousand seed weight was measured using a seed counter.

## 2.5. Chlorophyll Fluorescence and Chlorophyll Content Measurements

A fluorimeter (OS5p; Opti-Sciences, Inc., Hudson, NY, USA) was used to measure the efficiency of the photosynthetic apparatus. Prior to fluorescence measurements, the upper surface of three healthy leaves at the top of one plant from three randomly selected sites for each plot was covered with leaf clips for 30 min. Leaf fluorescence was then measured with a light pulse of 15,000 $\mu$mol $m^{-2}$ $s^{-1}$ at a wavelength of 660 nm The assessed parameter was maximum photosynthetic efficiency of PSII ($F_v/F_m$), which was calculated using the following formula: $F_v/F_m = (F_m - F_0)/F_m$, on the basis of the measured parameters: minimal fluorescence ($F_0$), maximal fluorescence ($F_m$), variable fluorescence ($F_v$) [23].Chlorophyll content meter (CCM-200plus; Opti-Sciences, USA) was used to estimate the chlorophyll content index (CCI) on the same leaves that were used for chlorophyll $\alpha$ fluorescence measurements. CCM-200plus measures the chlorophyll absorbance and calculates the chlorophyll content index, which is proportional to the concentration of chlorophyll in the sample.

## 2.6. Soil Sampling for Biochemical Analyzes

Soil samples collected from the arable layer (0–20 cm) were used as the research material for biochemical analyses. Each year they were collected at four terms: First term—at the plants' emergence (BBCH 5–10), Second term—at the plants' full growth (BBCH 35–40), third term—at the at the plants' florescence (BBCH 51–59), fourth term—after harvest.

Soil samples were taken from five places of each experimental plot, in four replications for each of the nine treatments of the experiment. In this way, at each analysis term we received 36 samples of soil, each of 1 kg.

## 2.7. Soil Enzymatic Activity

The analyses of soil enzymatic activity in individual treatments were based on the colorimetric method applied to measure the dehydrogenase activity (DHA), where 1% triphenyl tetrazolium chloride (TTC) was used as the substrate. The activity was measured after 24-h incubation at a temperature of 30 °C and a wavelength of 485 nm and it was expressed as $\mu$mol triphenyl formazane TPF 24 $h^{-1}$ $g^{-1}$dm of soil [24].

Apart from that, the biochemical analyses of soil involved the determination of activities of acid (EC 3.1.3.2) phosphomonoesterases (PAC) and alkaline phosphomonoesterases (PAL) with the method developed by Tabatabai and Bremner [25]. The activities were determined with disodium *p*-nitrophenyl phosphate tetrahydrate used as a substrate after 1 h incubation at 37 °C and at a wavelength of 400 nm. The results were converted into $\mu$mol (p-nitrophenol) PNP $h^{-1}$ $g^{-1}$dm of soil.

Catalase activity was measured by means of permanganometry, according to the method developed by Johnsons and Temple [26], where 0.3% $H_2O_2$ was the substrate. After 20-min incubation at room

temperature (about 20 °C) 0.02 M $KMnO_4$ was titrated to a light pink colour and expressed as µmol $H_2O_2$ $g^{-1}$ dm $min^{-1}$.

### 2.8. Biological Index of Fertility

The biological index of fertility (BIF) was calculated using the dehydrogenase activity (DHA) and catalase activity (CAT) according to the Stefanic method [27] using the following formula: (DHA + kCAT)/2, where k was the factor of proportionality which equaled 0.01.

### 2.9. Statistical Analyses

The dynamics of changes in the soil enzymatic activity was statistically analyzed. As there were no significant differences between the parameters in the research years, they were treated as replicates and the results were analyzed by two-way ANOVA using Statistica 12.0 software. The fertilization method and the term of analysis were the factors differentiating the traits under study to estimate the soil biochemical activity parameters. Homogeneous subsets of mean were identified via Duncan's test at a significance level of $p = 0.05$. Yield, biometric, physiological traits of plants, and nitrogenase activity were tested once a year for the experiment. Hence, one-way analysis of variance (ANOVA) was used with Duncan's confidence interval, which was applied at a significance level of $p = 0.05$. As there were no significant differences between the parameters in the experimental years, they were treated as replicas.

Principal component analysis (PCA) was used to visualize the multidimensional dependencies between the soil biochemical activity and the types of fertilization [28]. In order to show the existing regularities (correlations) between biometric and physiological parameters of plants in individual years of research, a Pearson correlation matrix was determined, which was illustrated using a heatmap. The colors indicate the correlation coefficient values (from darkest—value −1, to the lightest—value +1). Cluster analysis enables grouping of the studied physiological parameters of the plants in the experiment in such a way that the degree of correlation between parameters within one group was the highest and between groups the smallest [29]. The agglomeration Ward method (Ward Hierarchical Clustering) and the Euclidean distance were used to create a tree diagram.

## 3. Results

### 3.1. Yield, Biometric, and Physiological Traits of White Lupine Plants

The studied biostimulators/foliar fertilizers modified the yield and yield components of white lupine. The yield of white lupine seeds was low and ranged from 11.67 dt·$ha^{-1}$ (ADOB B) to 13.88 dt·$ha^{-1}$ (Optysil) and depended significantly on the bio-stimulants or foliar fertilizers that were applied (Table 3). After applying Optysil or ADOB Zn IDHA (13.63 dt·$ha^{-1}$), the yields were significantly higher when compared to the control plants by 1.82 and 1.57 dt·$ha^{-1}$, respectively.

Thousand seed weight (TSW) was significantly higher than the control plants when ADOB Zn IDHA (322.7 g) was applied. All tested preparations significantly stimulated the height of white lupine. The strongest stimulation was obtained by Metalosate potassium, which increased the height of white lupine (40.5 cm) by 6.2 cm when compared to the control. Apart from these fertilizers, in the group that most strongly stimulated this trait were: Optysil (39.8 cm), ADOB 2.0 Mo (38.9 cm), and Bolero Mo (38.6 cm).

ADOB Zn IDHA (318.4 pc·$m^{-2}$) and Tytanit (300.8 pc·$m^{-2}$) significantly increased the number of pods compared to the control, and the increase was 96.6 and 79 pc·$m^{-2}$, respectively.

Studies have also shown changes in nodulation and physiological parameters of the plant. Dry mass of root nodules was significantly stimulated after application of ADOB Zn IDHA (0.212 g) by 0.067 g when compared to the control treatment.

Chlorophyll fluorescence ($F_v/F_m$), showing the level of plant stress, was measured in the BBCH 69 (end of flowering) and BBCH 79 phases (75% of the pods reached typical length). At the end of flowering, the best plant condition, expressed by the $F_v/F_m$ parameter, was obtained after the

application of ADOB Zn IDHA (0.815) or Metalosate potassium (0.813) and both values significantly exceeded those obtained both with the control and all other treatments. In the assessment made at a later developmental phase, the tested biostimulators/foliar fertilizers did not significantly differentiate this parameter.

ADOB Zn IDHA application significantly stimulated the content of chlorophyll in leaves, expressed in CCI, which was 50.9 and exceeded the control by 17.4, as well as all other objects. In addition, significantly higher CCI values than in the control object were obtained after using Tytanit (46.7), ADOB B (42.9), or ADOB 2.0 Mo (40.7).

In turn, the significantly highest LAI value in the experiments was obtained after application of Rooter. The LAI value was 2.03 and exceeded the control by 0.62, for which the lowest LAI value was determined.

**Table 3.** The influence of the bio-stimulants and fertilizers on yield, biometric, and physiological traits of white lupine.

| Objects | Seed Yield, dt·ha$^{-1}$ | TSW, g | Height, cm | Number of Pods, pc.·m$^{-2}$ | Plant Dry Mass, g | Root Nodules Dry Mass, g | $F_v/F_m$ BBCH 69 | CCI | LAI |
|---|---|---|---|---|---|---|---|---|---|
| 1 | 12.06 bc | 302.4 bc | 34.3 e | 221.8 bc | 5.05 | 0.145 bc | 0.784 cd | 33.5 d | 1.41 g |
| 2 | 11.82 bc | 295.2 c | 37.8 bcd | 300.8 a | 5.72 | 0.147 bc | 0.796 b | 46.7 b | 1.73 de |
| 3 | 13.88 a | 301.7 bc | 39.8 ab | 250.5 b | 6.46 | 0.160 bc | 0.792 bc | 23.1 f | 1.81 c |
| 4 | 11.96 bc | 305.8 bc | 40.5 a | 189.1 c | 5.28 | 0.170 b | 0.813 a | 24.2 f | 1.70 de |
| 5 | 13.18 ab | 313.9 ab | 37.1 cd | 235.0 bc | 5.36 | 0.142 bc | 0.776 d | 25.6 f | 2.03 a |
| 6 | 12.76 abc | 306.1 bc | 38.6 abc | 250.0 b | 5.14 | 0.129 c | 0.774 d | 30.2 e | 1.88 b |
| 7 | 13.63 a | 322.7 a | 36.4 d | 318.4 a | 6.17 | 0.212 a | 0.815 a | 50.9 a | 1.61 f |
| 8 | 11.67 c | 310.4 b | 37.9 bcd | 273.4 ab | 5.12 | 0.169 bc | 0.779 d | 42.9 c | 1.68 e |
| 9 | 11.94 bc | 309.0 b | 38.9 abc | 240.7 b | 5.24 | 0.170 bc | 0.797 b | 40.7 c | 1.76 cd |
| *p*-value | 0.001 | 0.000 | 0.000 | 0.000 | 0.236 | 0.001 | 0.000 | 0.000 | 0.000 |

1. control—no bio-stimulants or foliar fertilizers applied to the plants; 2. plant + Tytanit; 3. plant + Optysil; 4. plant + Metalosate potassium; 5. plant + Rooter; 6. plant + Bolero Mo; 7. plant + ADOB Zn IDHA; 8. plant + ADOB B; 9. plant + ADOB 2.0 Mo; lack of homogeneous groups means no significant differences at the level of *p* < 0.05, a, b, c, d, e, f, g-homogeneous groups (Duncan's test. *p* < 0.05); TSW-thousand seed weight, $F_v/F_m$—maximum photosynthetic efficiency of PSII, CCI—chlorophyll content index, LAI—leaf area index.

The results of the experiment showed that foliar fertilizers and bio-stimulants affected the enzymatic activity of the soil and the biological index of fertility (BIF), as well as the nitrogenase activity in the white lupine plantation. The two-way analysis of variance showed that the foliar fertilization/bio-stimulants did not have a significant influence on the enzymatic activity and the soil biological index of fertility (BIF). Only the term of the test (development phase, based on BBCH scale) had a highly significant influence on the enzymatic activity and the biological index of fertility (BIF) of the soil (Table 4). One-way analysis of variance showed that foliar fertilization/bio-stimulants had a significant influence on nitrogenase activity.

**Table 4.** The test *F* statistics and the significance levels of the two-way analysis of variance for the soil bioactivity. The traits under analysis were affected by two factors, i.e., foliar fertilization and the term of the test.

| Parameter | Fertilization | Development Phase | Interaction |
|---|---|---|---|
| **White Lupine *Butan*** | | | |
| Dehydrogenase | 13.393 ns | 159.989 *** | 41.123 ns |
| Alkaline phosphatase | 7.036 ns | 51.672 *** | 5.37 ns |
| Acid phosphatase | 14.907 ns | 116.200 *** | 10.116 ns |
| Catalase | 192.47 ns | 1558.42 *** | 121.42 ns |
| BIF | 2.90 ns | 131.96 *** | 2.71 ns |
| Nitrogenase | 14.08 *** | - | - |

*F* test statistics and significance levels of two-way analysis of variance for activity of enzymes associated with herbicides and terms research fixed factors *** *p* = 0.001, ns—no signification.

### 3.2. Biological Fixation of Nitrogen under Lupine Plantation

The field analyses of the biological fixation of nitrogen showed that the fertilizers and bio-stimulants significantly stimulated the nitrogenase activity in the white lupine plantation (Figure 2). During the three years in all the experimental treatments nitrogenase exhibited higher activity than in the control plot and differences were statistically significant. The highest nitrogenase activity was noted after the application of the ADOB B and ADOB Zn IDHA. The activity of the enzyme was respectively six and four times higher than in the control plot. Apart from the control treatment, the lowest biological fixation of nitrogen was noted after the application of Metalosate potassium.

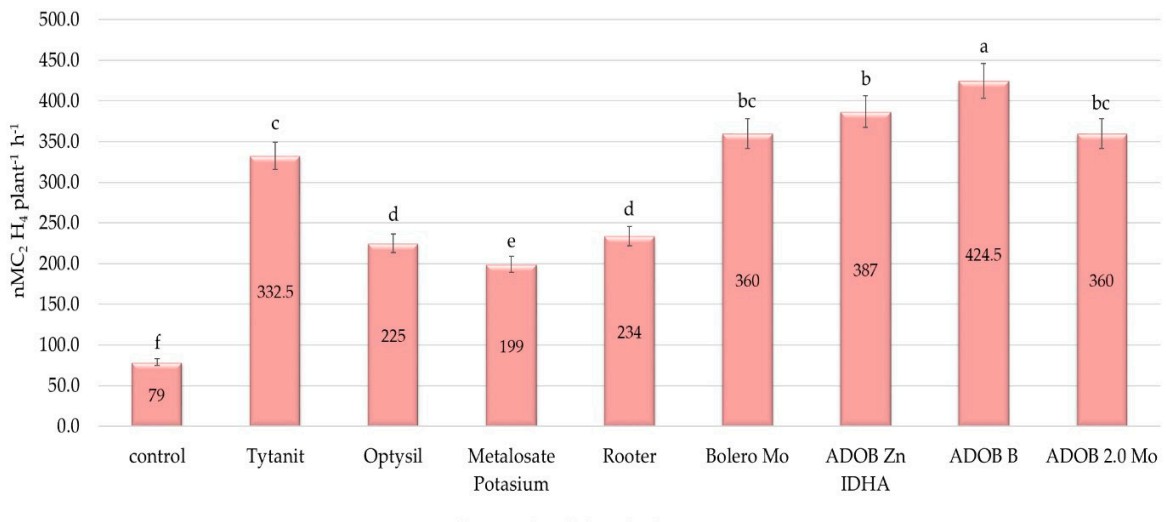

**Figure 2.** The influence of the bio-stimulants and fertilizers on the level of biological fixation of nitrogen. Abbreviation: means values ± standard errors; a, b, c, d, e, f—homogenous groups according to Duncan's test at level $p = 0.05$.

The heat map presents correlations between all biometric and physiological characteristics of white lupine plants studied (Figure 3). Based on this visualization, relatively higher correlations were found between some features, including: PN (number of pods, pc.·m$^{-2}$), TSW (thousand seed weight), H (height plant), PDM (plant dry mass), Y (seed yield), and PDM, Y, LAI (leaf area index) and Fv/Fm[1] (maximum photosynthetic efficiency of PSII BBCH–69). In turn, BNF (biological nitrogen fixation) and RNDM (root nodules dry mass) are negatively correlated with LAI, Y, PDM, H, TSW, PN, and Fv/Fm[2] (maximum photosynthetic efficiency of PSII BBCH–78). Additionally, based on cluster analysis, groups of related biometric and physiological traits of plants were determined. Three groups have been designated. The first group that is the most distinct from the others contains: RNDM, BNF, CCI, and Fv/Fm[1]. The other two groups are: LAI, Y, PDM and H, TSW, PN, Fv/Fm[2].

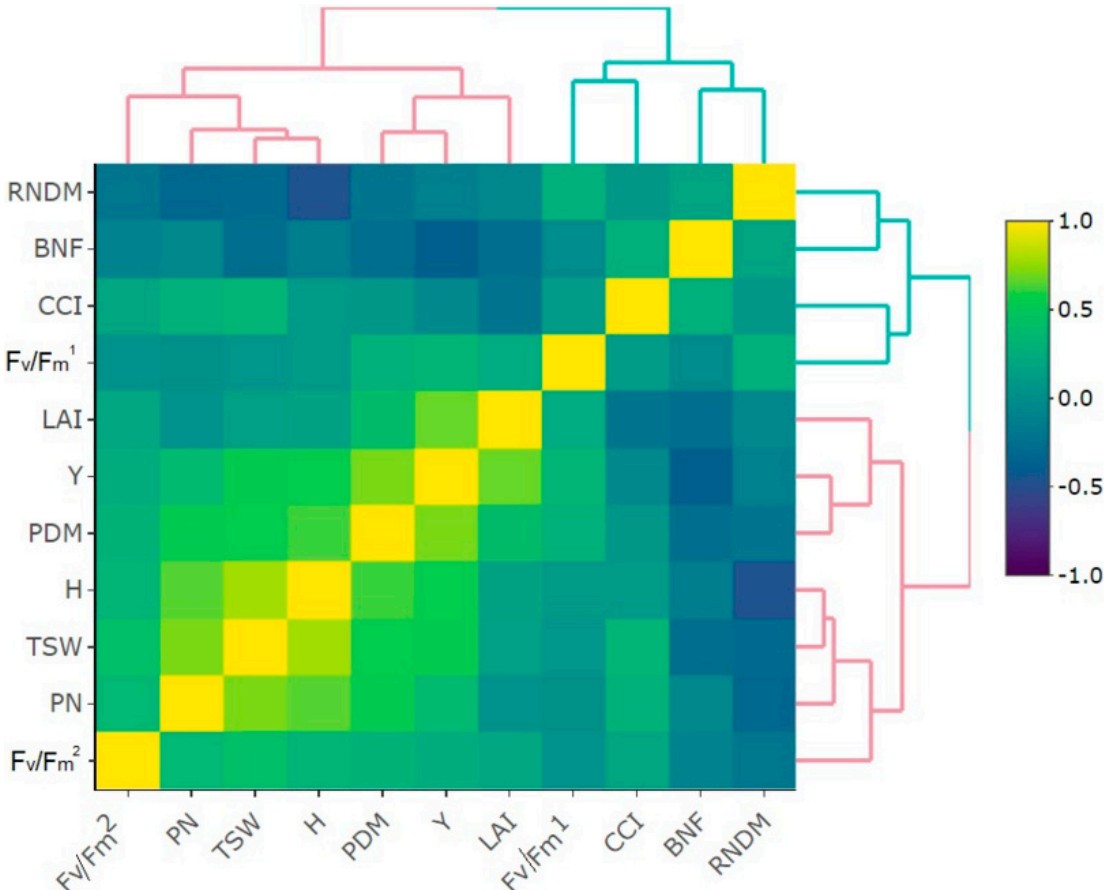

**Figure 3.** Correlations between all biometric and physiological characteristics of white lupine plants. Abbreviation: RNDM—root nodules dry mass, g—BNF—biological nitrogen fixation, CCI—chlorophyll content index, Fv/Fm$^1$—maximum photosynthetic efficiency of PSII BBCH–69, LAI—leaf area index, Y—seed yield, PDM—plant dry mass, g, H—height plant, TSW—thousand seed weight, PN—number of pods, pc.·m$^{-2}$; Fv/Fm$^2$—maximum photosynthetic efficiency of PSII BBCH–78.

### 3.3. Analysis of Soil Biochemical Activity

Only the ADOB 2.0 Mo and Metalosate potassium foliar fertilizers stimulated the dehydrogenase activity throughout the growing season, as compared with the control treatment. After the application of the bio-stimulants the level of the enzyme activity was similar to the activity in the control treatment. However, when the Optysil and ADOB B were applied, the activity decreased but not statistically significantly. The experiment also showed that the peak of the dehydrogenase activity significant occurred at the third term of analyses, when the plants began flowering (BBCH 51–59). The results of the analysis of the dehydrogenase activity in the soil under the white lupine plantation are shown in Figure 4.

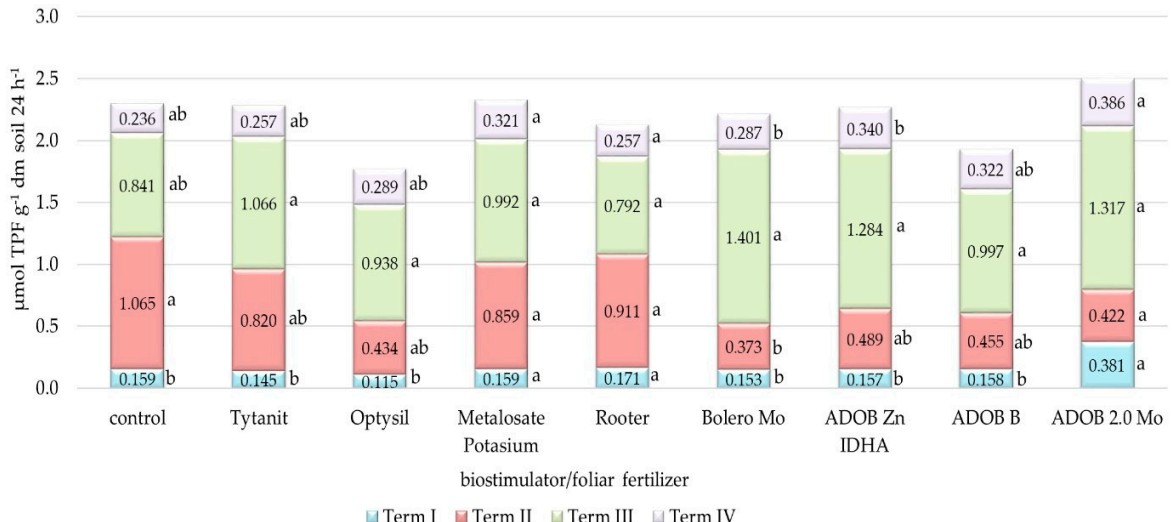

**Figure 4.** The influence of the bio-stimulants and fertilizers on the dehydrogenase activity. Abbreviation: a, b—homogenous groups according to Duncan's test at level $p = 0.05$; I term—at the plants' emergence (BBCH 5–10), II term—at the plants' full growth (BBCH 35–40), III term—at the at the plants' florescence (BBCH 51–59), IV term—after harvest.

The analysis of the results of the acid phosphatase activity (PAC) shows that during the entire white lupine growing season the foliar fertilizers and bio-stimulants decreased the activity of this enzyme, as compared with the control treatment (Figure 5). This effect was not observed when the Metalosate potassium foliar fertilizer was applied. During the second term of analyses, shortly before flowering, the acid phosphatase activity in all the experimental treatments was higher than in the control treatment. It was very high after the application of the Bolero Mo (0.170 μmol PNP h$^{-1}$ kg$^{-1}$dm of soil) and ADOB 2.0 Mo (0.171 μmol PNP h$^{-1}$ g$^{-1}$dm of soil).

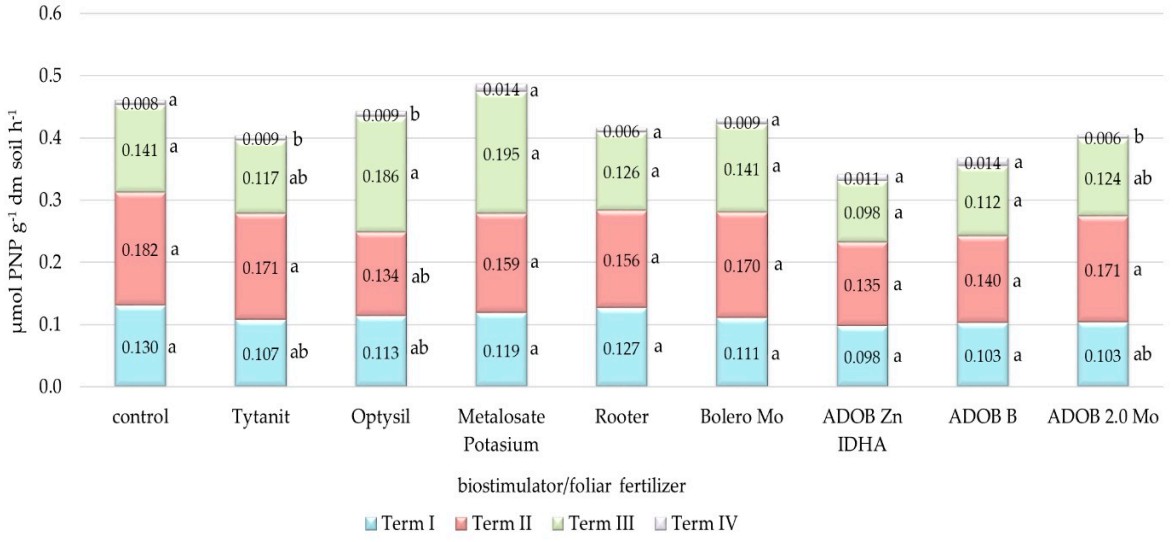

**Figure 5.** The influence of the bio-stimulants and fertilizers on the acid phosphatase level. Abbreviation: a, b—homogenous groups according to Duncan's test at level $p = 0.05$; I term—at the plants' emergence (BBCH 5–10), II term—at the plants' full growth (BBCH 35–40), III term—at the at the plants' florescence (BBCH 51–59), IV term—after harvest.

The bio-stimulants and most of the foliar fertilizers did not increase the alkaline phosphatase (PAL) activity in the white lupine plantation, as compared with the control treatment (Figure 6). The ADOB 2.0 Mo and Bolero Mo stimulated the activity of this enzyme, which respectively increased by

14% and 5%, as compared with the control treatment. The enzyme exhibited statistically significantly increased activity shortly before they began flowering (II term).

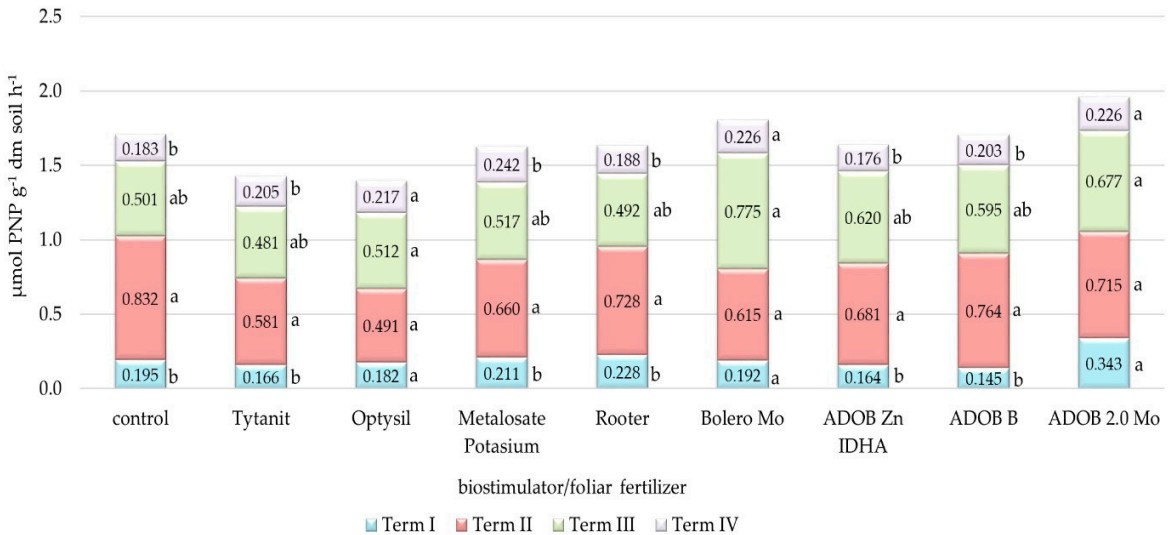

**Figure 6.** The influence of the bio-stimulants and fertilizers on the alkaline phosphatase level. Abbreviation: a, b—homogenous groups according to Duncan's test at level $p = 0.05$; I term—at the plants' emergence (BBCH 5–10), II term—at the plants' full growth (BBCH 35–40), III term—at the at the plants' florescence (BBCH 51–59), IV term—after harvest.

All the preparations stimulated the catalase activity, as compared with the control treatment (Figure 7), but not significantly. The enzyme significantly exhibited high activity, i.e., when the plants started flowering (III term) in all the experimental treatments. The catalase activity ranged from 98.510 $\mu$mol $H_2O_2 g^{-1}$ dm min$^{-1}$ after the application of the Tytanit to 135.819 $\mu$mol $H_2O_2 g^{-1}$ dm min$^{-1}$ after the application of the Bolero Mo.

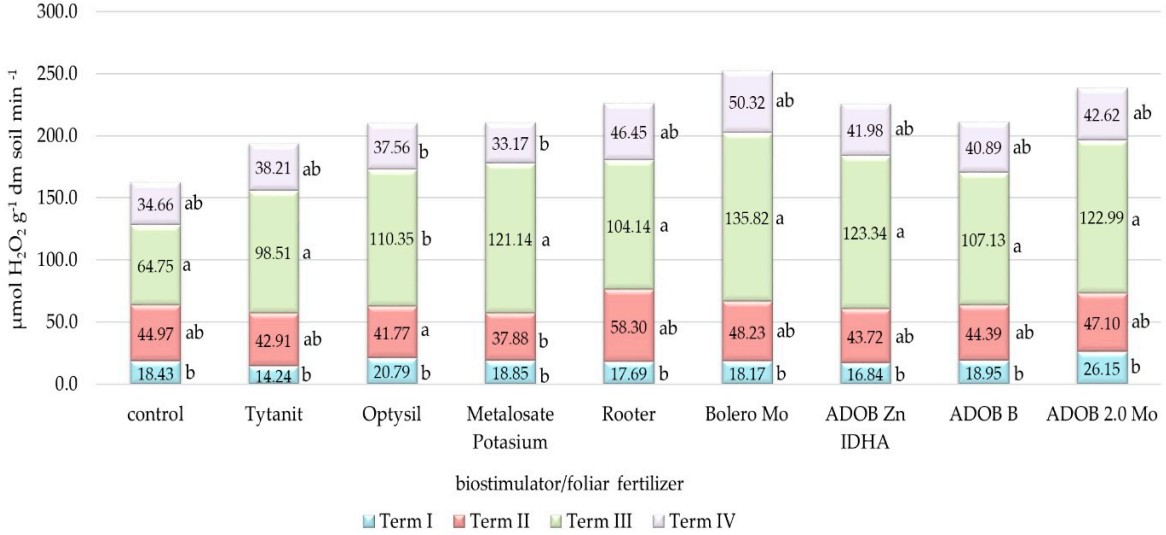

**Figure 7.** The influence of the bio-stimulants and fertilizers on the catalase activity. Abbreviation: a, b—homogenous groups according to Duncan's test at level $p = 0.05$; I term—at the plants' emergence (BBCH 5–10), II term—at the plants' full growth (BBCH 35–40), III term—at the at the plants' florescence (BBCH 51–59), IV term—after harvest.

The biological index of fertility (BIF), which was calculated on the basis of the dehydrogenase and catalase activity, was not always higher after the application of the bio-stimulants and foliar fertilizers

(Figure 8). The highest value of this indicator was noted after the application of the Optysil and the lowest after ADOB Zn IDHA. The BIF was significantly high at the beginning of flowering, as it ranged from 5.17 after the application of ADOB Zn IDHA to 12.34 after the application of ADOB 2.0 Mo. The indicator was also high after the application of the Bolero Mo and Optysil.

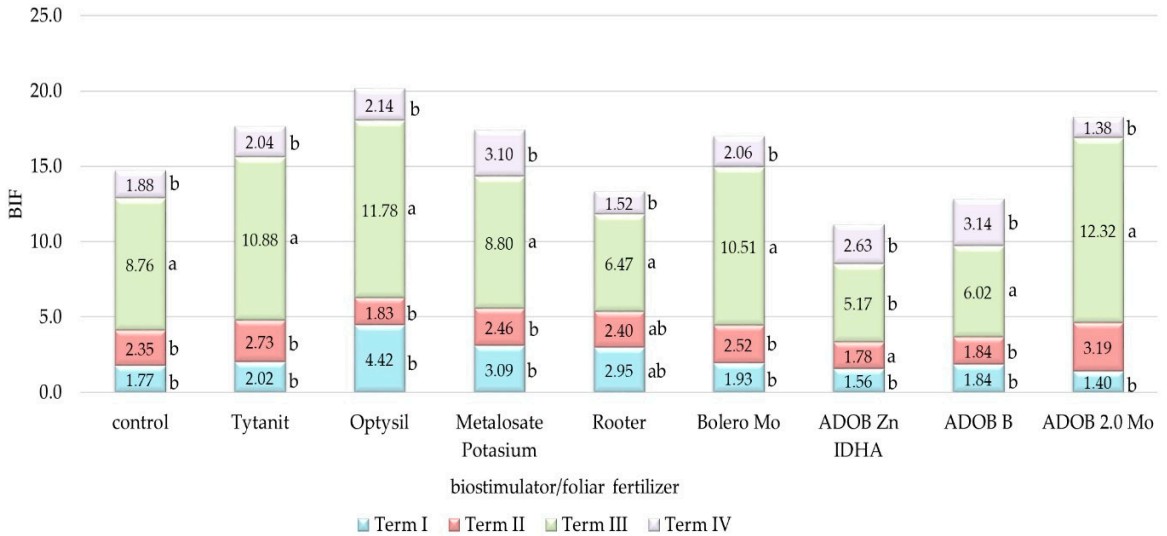

**Figure 8.** The influence of the bio-stimulants and fertilizers on the BIF. Abbreviation: a, b—homogenous groups according to Duncan's test at level *p* = 0.05; I term—at the plants' emergence (BBCH 5–10), II term—at the plants' full growth (BBCH 35–40), III term—at the at the plants' florescence (BBCH 51–59), IV term—after harvest.

Principal component analysis (PCA) was used to show how the foliar fertilizers and bio-stimulants affected the white lupine plantation. The first two principal components accounted for over 89.2% of the total variation (Figure 9). The parameters of the soil biochemical activity in 2018 differed significantly from 2016 to 2017. This effect may have been caused by the weather conditions (Figure 1). In 2018 the season was the warmest of all the research years. The average temperature difference between 2018 and the previous years was 2.9 °C in August and 1.7 °C in May. As the thermal conditions were very similar in 2016 and 2017, the PCA showed similar dependencies for these two years. In 2016 the fertilizer preparations and bio-stimulants significantly affected the catalytic activity of acid phosphatase (PAC) at all the terms of analyses. This dependency was not observed for the other parameters of soil biochemical activity.

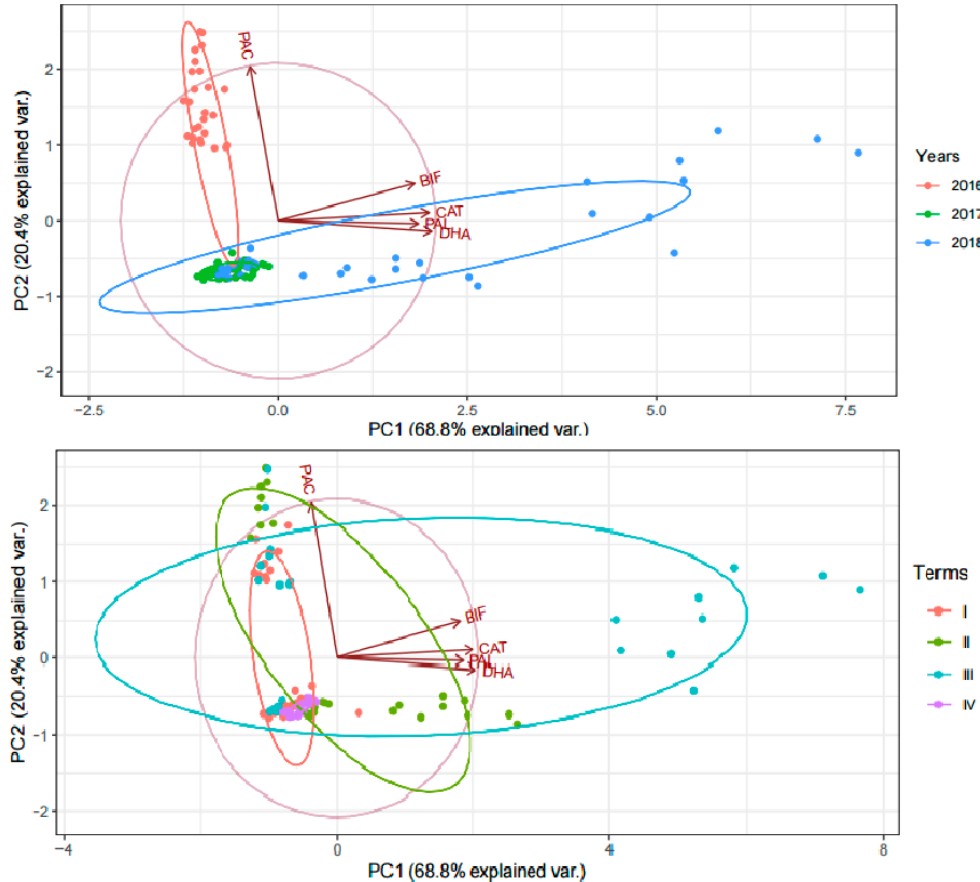

**Figure 9.** The dependence between the soil enzymatic activity and all treatments with fertilizers and bio-stimulants at the terms of analyses. Abbreviation: I term—at the plants' emergence (BBCH 5–10), II term—at the plants' full growth (BBCH 35–40), III term—at the at the plants' florescence (BBCH 51–59), IV term—after harvest. BIF—index of fertility, CAT—catalase activity, PAC—acid phosphomonoesterases, PAL—alkaline phosphomonoesterases, DHA—dehydrogenase activity.

In 2017 the application of the fertilizers did not cause significant differences in the activity of the soil enzymes or the biological index of soil fertility. In dry 2018 the preparations did not significantly affect the catalytic activity of the test parameters only during plants' emergence (I term). However, at the plants' full growth (II term), the foliar fertilizers and bio-stimulants strongly influenced the catalytic activity of catalase (CAT), dehydrogenase (DHA), alkaline phosphatase (PAL), and the biological index of soil fertility (BIF). Apart from that, the principal component analysis showed that in 2018 the indicators of soil biochemical activity were affected most strongly by foliar fertilizers and bio-stimulants the flowering of the plants (III term) and after the harvest (IV term).

## 4. Discussion

### 4.1. Yield, Biometric, and Physiological Traits

Silicon, iron, manganese, boron, copper, molybdenum, and zinc are the basic micronutrients. The silicon content in most plants is comparable to the content of calcium, magnesium, and phosphorus. Many studies have shown the positive effects of silicon on plants, their development, yield, and sensitivity to biotic and abiotic stress [30]. In many tests, silicon has been shown to significantly influence the regulation of nutrient uptake such as: calcium, magnesium, and phosphorus. In other studies [31], silicon fertilization increased the yield of sugar beet roots by 13.7–15.9%, as well as the yield of many other species [11], especially in the form of spraying plants under stress conditions. According to Fageria and Baligar [32] and Duffy [33] Zn is the microelement most limiting crop yield.

Zinc is taken up in small amounts and it participates in all major functions of the plant, increases nitrogen uptake, and activates $CO_2$ binding in later stages. Hence, it is necessary in plant nutrition and its importance in plant production is growing [13]. Similarly, Kaya et al. [34] obtained the highest common bean plants (*Phaseolus vulgaris* L.), with the largest number of pods and seeds per plant after application of a foliar mixture of zinc.

The preparations used in our study also stimulated the tested biometric parameters of the plants. Plant height was stimulated the most after application of Metalosate potassium (by 8.5%) when compared to the control treatments. In turn, the number of nodules was most strongly stimulated by ADOB Zn IDHA (by 68.4%) and LAI by Rooter (by 69.5%). Other fertilizers containing boron, molybdenum, silicon, and titanite also increased the parameters indicated above. These results are consistent with the results of Raj and Raj [12] regarding the beneficial effects of Zn on plant efficiency, physiological parameters, plant height, and nodulation formation. Our results are also consistent with field studies of Omer et al. [35], in which the treatments of molybdenum application did not modify any of the studied lentil characteristics, except for the height of the plant. Also Rahman et al. [36] showed that the use of molybdenum in its deficiency in soil, stimulates the formation of nodules. Of the physiological traits studied, chlorophyll fluorescence (Fv/Fm) was most strongly stimulated by ADOB Zn IDHA (by 3.9%) and Metalosate potassium (by 3.7%). In turn, the CCI index was most strongly stimulated by ADOB Zn IDHA, whose application resulted in an increase of this parameter by 51.9% when compared to the control treatment. The results of research on *Vigna sinensis* [37] and on *Celosia* [38] showed that Zn spraying on plants caused a significant increase in chlorophyll content. In a study conducted by Artyszak et al. [39], foliar fertilization with silicon increased LAI and effective quantum efficiency of PSII—ΦPSII, as well as positively affected the growth and development of many plant species [40,41].

## 4.2. Biological Fixation of Nitrogen

The bio-stimulants and foliar fertilizers which improved the biological fixation of nitrogen in the white lupine plantation contained important macro- or microelements. Scientific reports suggest that some elements are particularly significant to the nitrogen fixation process.

Mineral nutrients may influence $N_2$ fixation in legumes at various stages of the symbiotic process: infection and nodule development, nodule function, and host plant growth. For healthy and vigorous growth, intact plants need to take up relatively large amounts of some inorganic elements: ions of nitrogen (N), potassium (K), calcium (Ca), phosphorus (P) and sulphur (S), and small quantities of other elements: iron (Fe), nickel (Ni), chlorine (Cl), manganese (Mn), zinc (Zn), boron (B), copper (Cu), and molybdenum (Mo). Molybdenum and iron are especially important because they are components of the nitrogenase complex in rhizobia which is required for nitrogen fixation. They are components of nitrogenase—the bacterial enzyme that enables the diazotrophy process. The nitrogenase protein consists of two subunits: the larger one containing the FeMo cofactor and the smaller one containing iron alone [42]. Plants growing on acidic, moist, and poorly buffered soils do not have sufficient supply of molybdenum. When molybdenum is applied in a field to the leaves of legumes, the nitrogen fixation of these plants is more efficient, and the mass of their root nodules and the yield of seeds increase [43,44]. The use of ADOB 2.0 Mo with high molybdenum content in our experiment confirmed this fact. There are small amounts of boron in plants, but this micronutrient plays an important role in various physiological processes. It affects the separation of plant tissues and it is necessary for the optimal growth of plants. Boron-deficient plants have less bacteria of the *Rhizobium* genus and fewer infection threads [44]. The significant increase in the level of biological fixation of nitrogen may have been caused by the application of the foliar fertilizer containing boron (ADOB B). Our research also proved that zinc supplied with the ADOB Zn IDHA foliar fertilizer significantly increased the nitrogenase activity. Although plants absorb moderate amounts of zinc, this element has significant influence on bacteria of the *Rhizobium* genus. The research by Wani et al. [45] showed that higher concentrations of this element in soil stimulate bacteria of the *Rhizobium* genus to produce phytohormones (including

indoleacetic acid), which promote the growth of plants by increasing the number of root nodules, their dry mass, and the content of leghemoglobin in the nodules.

Many researchers have studied the role of phosphorus in symbiotic systems. Phosphorus plays a crucial role in the nitrogen fixation process [46,47]. The Rooter bio-stimulant, which contained phosphorus and potassium, stimulated this process considerably. Phosphorus participates in a wide range of molecular and biochemical processes. Apart from that, some phosphate bonds are carriers of the energy used in cells. The presence of phosphorus in soil affects the plant's ability to produce root nodules, especially the weight and the number of nodules [48], which translates into the level of nitrogen fixation.

When the supply of phosphorus is insufficient, plants often suffer from nitrogen deficiency. Sulphur and potassium are less important for symbiotic systems than the aforementioned elements. Nevertheless, potassium ions are very desirable in saline soils because they function as an osmolyte. In view of the fact that nearly half of irrigated soils around the world are considered saline, the addition of potassium helps to maintain the bacteria-plant system [48,49].

## 4.3. Biochemical Activity

The activities of soil enzymes are considered sensitive indicators of important microbial reactions involved in nutrient cycles and they respond to changes in the soil caused by natural or anthropogenic factors. In this regard, soil enzyme activities are often used to evaluate the impact of human activity on soil life [50].

Soil enzymes are a group of catalysts that significantly affect the ecological properties of the pedosphere. These are both extracellular enzymes and the ones that are present in microorganisms (both in proliferating cells and in endospores). Enzymes control the course of all chemical reactions in microbial cells, e.g., the synthesis of proteins, nucleic acids, and carbohydrates [51]. Soil enzymes are involved in the decomposition of organic substances released into the soil during the plant's growth as well as the formation and decomposition of humus in the soil. They release and transfer minerals to plants. In spite of the dynamic nature of the microbiological and biochemical properties, soil enzymes are accurate and significant determinants of soil fertility, and they are important indicators of changes taking place in the soil [52,53].

Dehydrogenases (DHA) are enzymes belonging to the group of oxidoreductases. They are responsible for catalyzing the oxidation of organic compounds. Active dehydrogenases are present only inside living cells and they indicate the presence of physiologically active microorganisms. Dehydrogenases are commonly found in the pedosphere, where they are involved in the decomposition of organic compounds. Measurement of the dehydrogenase activity in soil shows the intensity of respiratory metabolism of soil microorganisms, mainly actinobacteria and bacteria.

Our research showed that only some foliar fertilizers (ADOB 2.0 Mo and Metalosate Potassium) stimulated the dehydrogenase activity in the white lupine plantation, however, the results were not significant.

Dehydrogenase exhibited high activity at the beginning of the plants' flowering phase (BBCH 51–59). It may have been caused by an increased secretion from the root system during that period [54,55]. In consequence, the count of microorganisms increased [56].

Also macro- and microelements applied in the form of foliar fertilizers and biostimulators could affect dehydrogenase activity. Bielińska et al. [57] observed that fertilizing preparations with nitrogen, phosphorus, and potassium increased the content of these enzymes in the soil. There was a similar effect observed in our study after the application of the Metalosate potassium foliar fertilizer. There were analogous results of experiments on similar bio-conditioners conducted by [58] and [53]. According to Bilen et al. [59], boron improves the dehydrogenase activity. Taran et al. [60] showed that molybdenum stimulated the production of these enzymes by the root nodules of legumes. They also observed that the content of titanium might be positively correlated with the soil biochemistry.

The results of the experiment showed that both the bio-stimulants (Tytanit and Rooter) and foliar fertilizers positively affected the acid phosphatase activity, which was lower than in the control treatment. The Metalosate potassium foliar fertilizer did not cause this effect. This shows that the preparations used in our experiment positively influenced the plants' ability to absorb phosphorus. It is necessary to remember that phosphorus-deficient plants are characterized by increased secretion of acid phosphatase through the root system into the soil. Ciereszko et al. [61] found that the deficit of this macroelement stimulated plants' secretion of acid phosphatases. Lemanowicz et al. [62] and Niewiadomska et al. [56] also suggest these relationships in their studies on the effect of the PRP SOL fertilizer containing phosphorus, potassium, zinc, boron, and molybdenum on the lupine plantation. They observed a decrease in the catalytic activity of this enzyme because of the activation of the compounds that were inaccessible to plants. Bielińska and Mocek-Płóciniak [63] made similar observations. Wang et al. [64] also found that these enzymes exhibited higher activity in the experimental treatment without phosphorus fertilization.

The alkaline phosphatase activity increased significantly only after the application of the ADOB 2.0 Mo and Bolero Mo foliar fertilizers. This effect may have been caused by the increased activity of soil microorganisms, which were stimulated by organic phosphorus compounds secreted into the soil by white lupine plants. Waldrip et al. [65] proved that the content of organic forms of phosphorus was correlated with the activity of alkaline phosphatases in the soil.

All the preparations used in the experiment significantly stimulated catalase activity. As early as 1963, Koter [66] found that the catalase activity increased when plants were fertilized with boron. Hu and Zhu [67] observed that the catalase and dehydrogenase activity increased when plants were fertilized with silicon. Such elements as copper and zinc are essential constituents of physiological processes in all living organisms, including microorganisms. Some soils suffer from zinc deficits, which is why they are enriched with fertilizers containing this element to satisfy the nutritional requirements of crops and improved soil activity [68].

The results of the enzymatic analyses of the dehydrogenase and catalase activities enabled the calculation of the biological index of soil fertility (BIF). The treatments with the Optysil and ADOBE 2.0 Mo preparations had influence on the BIF values, as compared with the control sample. The use of the Optysil preparation resulted in particularly high values in the soil samples collected at the beginning of the flowering phase. The BIF value resulted from the significant influence of these fertilizers on the activity of catalase and dehydrogenase. Siwik-Ziomek and Szczepanek [69] indicated that mineral fertilization, which increases the yield of crops, indirectly affects root secretion, and thus increases the biochemical activity of soil at specific phases of plants' development.

## 5. Conclusions

When non-root fertilization is applied to plants, they take up all necessary elements chiefly through their leaves as well as the stalk and the whole aerial system. A strong stimulating effect on the yield of white lupine plants in comparison with the control object was obtained after the application of silicon (Optysil) or chelated zinc (ADOB Zn IDHA). The use of zinc in foliar fertilizers (ADOB Zn IDHA) in comparison with control treatment stimulated most of the tested features/parameters: TSW, number of pods per 1 $m^2$, root nodules dry matter, photochemical efficiency of PSII ($F_v/F_m$), and chlorophyll content (CCI). However, it is noteworthy that this way of "feeding" cannot substitute soil fertilization. It can only be used to quickly supply necessary nutrients to plants at difficult phases so as not to slow down their growth. The bio-stimulants and foliar fertilizers used in our study improved some of the biochemical parameters of soil activity and the nitrogen fixation process in the white lupine plantation. This effect may have been caused by the higher rate of penetration and better uptake of nutrients applied to the plants' leaves. Although macro- and micronutrients differ in their penetration rates, this process can be accelerated up to about a dozen times by non-root fertilization. The downside of foliar fertilization is the fact that only a limited amount of fertilizer can be supplied to plants in this way. Therefore, this method is particularly effective when plants need to be provided with the elements

they need in smaller amounts, e.g., iron, boron, and molybdenum. Not only is foliar fertilization a more efficient method of supplying micronutrients, but it is also safer for the environment and the plants. The search for methods that improve the yield and biochemical parameters of the soil environment is in agreement with the sustainable agriculture policy.

**Author Contributions:** A.N., H.S., A.W.-M., K.R., and Z.W. conceived and designed the experiments; A.N., H.S., A.W.-M., K.R., and Z.W. performed the field experiments and analyzed the data; A.B. statistical analysis; A.N., Z.W., H.S. wrote the paper; A.N., Z.W. revised the manuscript. All authors have read and agreed to the published version of the manuscript.

**Funding:** This research was funded by the Ministry of Agriculture and Rural Development of Poland, (it was not a grant. And The APC was funded by framework of Ministry of Science and Higher Education programme Project No. 005/RID/2018/19.

**Acknowledgments:** This research was financed by the Ministry of Agriculture and Rural Development of the Republic of Poland in the framework of the Multi-annual Program "Increasing the use of domestic feed protein for the production of high quality animal products in conditions of balanced development" implemented at the Department of Agronomy of the University of Life Sciences in Poznan. The publication was co-financed within the framework of Ministry of Science and Higher Education programme as, Regional Initiative Excellence" in years 2019-2022, Project No. 005/RID/2018/19.

**Conflicts of Interest:** The authors declare no conflict of interest.

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
