# Peer review of "The Influence of Bio-Stimulants and Foliar Fertilizers on Yield, Plant Features, and the Level of Soil Biochemical Activity in White Lupine (Lupinus albus L.) Cultivation"

_agronomy, doi:10.3390/agronomy10010150_

Round 1
Reviewer 1 Report
Authors have properly addressed all the comments from reviewers.
Author Response
Authors would like to thank for all valuable advices. This work language was corrected by native speaker.

Reviewer 2 Report
Thank you for sharing your work. I enjoyed reading about research outside of my specific focus but enjoyed seeing more applied research related to my area of expertise.
I have included many minor comments throughout the attached PDF file. In addition, below I outline some major comments:
Overall, the study was thorough and the methods appropriate, but the presentation and discussion of the results must be improved. Please seek to use a consistent format when preparing the figures and provide more detailed figure captions. Also, seek to be more concise in your writing focusing on describing the most important results and concepts in a logical flow. Please also avoid redundancy in describing the results.
As described in the manuscript, many of your findings agreed with those of past studies. To better highlight this, you need to organize your discussion in a way that your results are directly linked with past studies within the same paragraph. In its present form, the manuscript contains excessive paragraph breaks which make it difficult as a reader to link related information.
Be careful of stating that your results or those of others you include in your discussion prove something. Rather, use statements like indicate or suggest. It is dangerous to assume that results that correlate with an outcome are proof of the cause of an observed outcome. Although they may very well be, as scientists we remain skeptical so we can remain open-minded to all possibilities.
Please seek to address these major comments and the minor comments in order to improve your manuscript.

Author Response
Please find enclosed our revised manuscript entitled (changed according reviewer suggestion): "The Influence of Biostimulants And Foliar Fertilizers on Yield, Plant Features and the Level of Soil Biochemical Activity in White Lupine (Lupinus albus L.) Cultivation", which I am a corresponding author.
This work language was corrected by native speaker.
We have revised manuscript according reviewer’s suggestions and we would like to mention that all suggestions were taken into consideration.
Please find below the list of changes made in the manuscript according the reviewer’s suggestions.
We also would like to thank for all valuable advices, which improve our manuscript and make it more scientific valuable.
List of changes in the manuscript:
All of the minor comments included in PDF file were improved according to instructions. The presentation and discussion of the results were corrected according to suggestions. The figures were improved and detailed figure captions were provided. The text was rewritten in the results and discussion sections according to Reviewer suggestions focusing on describing the most important results and concepts in a logical flow. Redundancy in describing the results and discussion was deleted. Discussion was organized in a way that our results are now directly linked with past studies within the same paragraph. Excessive paragraph breaks were deleted. Statements like ‘indicate’ or ‘suggest’ were used rather than certainty.

This manuscript is a resubmission of an earlier submission. The following is a list of the peer review reports and author responses from that submission.
Round 1
Reviewer 1 Report
Comments to the Author: This manuscript described about the influence of bio-stimulants and fertilizers for foliar application on the biological nitrogen fixation and soil biochemical activity (dehydrogenase, phosphatase and catalase) in white lupin. I understand that this manuscript (this study) important in the understanding and development of cultivation method of legume plant utilizing biological nitrogen fixation and soil biochemical activity. However, I think this manuscript has some problem that need correcting. My specific comments are as follow; 1) P1, L2; Foliar Fertilisers → Foliar Fertilizer
2) P1, L21; …and the index of soil fertility (BIF) → and the biological index soil fertility?
3) P3, L96; What is “nitragina”? “nitragina” is biofertilizer including Bradyrhizobium?
4) P3, L101; Author should describe about amount of base fertilizer.
5) P3, L105; Author should indicate soil characteristics such as soil pH, EC, total N, total P, and so on.
6) P4, L129; Did you use which part of plant when you measured ARA? Please describe details.
7) P5, L134; It is it correct this title? I think this content are different.
8) P5, L158; The calculation of BIF was used DHA and CAT. Why don’t you use the value of phosphatase activity?
9) P5, L169 and 172; Both BIF is same mean?
10) P6, Table 3; Fertilisation → Fertilization
11) P6, Table 3; I think you had better add about influence of annual data.
12) P6, Figure 2; The unit of ARA is not English. “roslina” → “plant”
13) P6, Figure 2; I think you had better change abscissa axis and vertical axis and add error bar.
14) P6, Figure 2; Did you investigate about plant growth such as plant height, plant dry weight, nodule number and so on? If you have these data, you should indicate that. Especially, nodule number and nodule dry weight should indicate with the data of ARA.
15) Figure 3, 4, 5, 6, and 7; I think you had better unify these figures as table, because these figures are hard for me to understand.
16) In the explain and data of this manuscript, I think that the effect of soil biochemical activity on plant growth do not indicate. Author should indicate the data of plant growth characteristics and amount of each element in white lupin.
Reviewer 2 Report
Manuscript ID: Agronomy-626845
The Influence of Biostimulants And Foliar Fertilisers on the Process of Biological Nitrogen Fixation and the Level of Soil Biochemical Activity in Lupine (Lupinus Albus L.) Cultivation
Alicja Niewiadomska , Hanna Sulewska, Agnieszka Wolna-Maruwka1 Karolina Ratajczak, Zyta Waraczewska, Anna Budka
The objective of this study was to evaluate the effect of selected biostimulators (Titanit, Rooter) and foliar fertilizers (Optysil, Metalosate Potasium, Bolero Bo, ADOB 2.0 Zn IDHA, ADOB B, ADOB 2.0 Mo) on the levels of activity of dehydrogenase, acid and alkaline phosphatase, catalase and the level of biological nitrogen fixation of white lupine cultivation. The study (field trial) was conducted for three years at the Gorzyn Experimental and Education Station, Poznan, University of Life Sciences in Poland. Based on the 3-year accumulated data, the activities of dehydrogenase were significantly stimulated by ADOB 2.0 and Metalose Potassium; alkaline phosphatase activity was increased significantly by ADOB 2.0 and Bolero Mo foliar fertilizers and soil fertility index was highly influenced by Optysil foliar fertilizer. In the case of acid phosphatase activity, significant decrease was observed on all the treatments except to the plants applied with Metalose potassium foliar fertilizer. The analysis of the levels of the biological nitrogen fixation showed that the foliar fertilizers and biostimulants has significant effect on the nitrogenase activity on the white lupine plantation. The authors concluded and suggested that foliar fertilization is an effective method in providing plants with micro elements such as iron, boron, and molybdenum. According to the authors, foliar fertilization is also safer for the environment and plants.
Major comments:
This research deals with important aspects in application of biostimulants and foliar fertilizers on soil biochemical and microbial activities. The results obtained provide useful insights into practical use of these technologies. However, this research contains several serious flaws in experimental design, methods and statistical analysis as described below. Authors should work on these points and revise manuscript for publication.
Specific comments:
Methodology:
Lines 95 and 96: specific method of Bradyrhizobium inoculation was not included in the paper. Lines 103 to 106: The soil classification was described however the authors did not include the soil chemical properties of the 3-year experiment. Was it not necessary for the experiment? The general methods and practice of white lupine cultivation was not mentioned properly. In lines 109-110 it was briefly mentioned, but there was no reference.
Results:
Lines 170-172. The authors were discussing about the significant influence of the foliar fertilization/biostimulant and the term of the test, werethe authors discussing about the individual effects of these factors or were the authors referring to the interaction effect of these two factors that had significant contribution on the enzymatic activity and biological index of fertility of the soil?
Table 3, ANOVA of the interaction of the foliar fertilization and the term of the test. Why did the authors decide to have this analysis, given that the foliar fertilizers they used have different terms of application? Also, their samples were from soil under the lupine plantation, which leads to my question on why did the authors used p = 0.001 and not the commonly used p = 0.05 or 0.01?
Line 176 and 177, the authors mentioned about the activity of enzymes associated with herbicide and terms research F test statistic and ANOVA. This analysis was not mentioned in the methodology. Also, it was not explained properly in the results.
Line 178-193. Biological fixation of nitrogen under lupine plantation, the authors claim that nitrogenase activity was significantly influenced by the fertilizers and biostimulants, however there was no statistical analysis to back this claim. The authors reported that the lowest nitrogenase activity was observed on the application of Bolero Mo, but in the figure, it was the application of Metalosate Potassium that had the lowest nitrogenase activity.
Figures 3-7, discussions of these items by the authors claim significance of the treatments from the control, however statistical analysis to back the significance claim is also not present. Also, it would be better if the authors included what term the fertilizers and biostimulants were applied in the graph, given that those have different timing of application. It would be much easier to read and understand the paper.
Lines 336-341. The authors discussed about the high activity of Dehydrogenase at the beginning of the plants’ flowering phase, they suspected that it may be caused by the increased secretion of the root system that triggered the microbial count and increased the dehydrogenase activity. They also hypothesized that this increase in dehydrogenase activity may be caused by the higher rainfall during the months of June. In their weather data, during the years of 2016 and 2017 the high rainfall was reflected there and in year 2018, there were drought in the months of May, June, and August. This can be used to differentiate and better discuss on what factor caused the high dehydrogenase activity of each year because in the authors’ discussion, they discussed it as an accumulated mean of the dehydrogenase activity of the 3 year experiment.